# Brain-derived neurotrophic factor genetic polymorphism rs6265 and creativity

**Elisabeth Hertenstein**[1,2]*, **Marion Kuhn**[3], **Nina Landmann**[3], **Jonathan-Gabriel Maier**[2], **Carlotta Louisa Schneider**[2], **Kristoffer Daniel Fehér**[1], **Lukas Frase**[3,4], **Dieter Riemann**[3], **Bernd Feige**[3], **Christoph Nissen**[1,5]

**1** Faculty of Medicine, Department of Psychiatry, University of Geneva, Geneva, Switzerland, **2** University Hospital of Psychiatry and Psychotherapy, University of Bern, Bern, Switzerland, **3** Faculty of Medicine, Department of Psychiatry and Psychotherapy, Medical Center–University of Freiburg, University of Freiburg, Freiburg, Germany, **4** Faculty of Medicine, Department of Psychosomatic Medicine and Psychotherapy, Medical Center–University of Freiburg, University of Freiburg, Freiburg, Germany, **5** Division of Psychiatric Specialties, Department of Psychiatry, Geneva University Hospitals (HUG), Geneva, Switzerland

* elisabeth.hertenstein@unige.ch

## Abstract

The protein brain-derived neurotrophic factor (BDNF) promotes neural plasticity of the central nervous system and plays an important role for learning and memory. A single nucleotide polymorphism (rs6265) at position 66 in the pro-region of the human BDNF gene, resulting in a substitution of the amino acid valine (val) with methionine (met), leads to attenuated BDNF secretion and has been associated with reduced neurocognitive function. Inhomogeneous results have been found regarding the effect of the BDNF genotype on behavior. We determined the BDNF genotype and performance on the Compound Remote Associate (CRA) task as a common measure of creativity in 76 healthy university students. In our main analyses, we did not find significant differences between met-carriers (n = 30) and non-met carriers (n = 46). In a secondary analysis, we found that met-carriers had a slower solution time (medium effect size) for items of medium difficulty. Our results suggest that met-carriers and non-met-carriers do not generally differ regarding their creativity, but non-met-carriers may have a certain advantage when it comes to moderately difficult problems. The wider literature suggests that both genetic variants come with advantages and disadvantages. Future research needs to sharpen our understanding of the disadvantages and, potentially, advantages met allele carriers may have.

**Data Availability Statement:** Upon consultation with the person responsible for data protection at the University of Freiburg, Germany, our data contain potentially identifying and sensitive

## Introduction

The protein brain-derived neurotrophic factor (BDNF) is a neurotrophin found in all mammals. BDNF is released following neuronal activity. Its main functions include neuroprotection, the stimulation of neuronal proliferation, and the modulation of synaptic interactions [1]. Since BDNF promotes long-term potentiation (LTP) and neural plasticity of the central nervous system, it plays an important role in learning and memory [1]. BDNF is widely accepted as an influential factor in the response to stress and the development of neuropsychiatric diseases and is a potential target for the development of new drugs [2]. BDNF is hypothesized to be an important mediator of treatment effects in patients with depression, because increasing

information. Data requests may be sent to the local ethics committee of the University of Freiburg, Germany: Albert-Ludwigs-Universität Freiburg, Ethik-Kommission, Engelbergerstrasse 21, 79106 Freiburg im Breisgau, Germany. Phone: +49 761 270 72640. Email: ekfr@uniklinik-freiburg.de.

**Funding:** The author(s) received no specific funding for this work.

**Competing interests:** The authors report there are no competing interests to declare.

neuronal plasticity might represent a central mechanism of action of many antidepressant strategies, including pharmacotherapy, electroconvulsive therapy, and psychotherapy [3, 4].

A naturally occurring single nucleotide polymorphism (rs6265) in the pro-region of the human BDNF gene, at position 66, resulting in a substitution of the amino acid valine (val) with methionine (met), is responsible for the presence of three different genotypes in humans: val66val (val/val), val66met (val/met) and met66met (met/met). The presence of the met-allele results in impaired BDNF secretion following neuronal stimulation and has been associated with reduced synaptic plasticity [5, 6], impaired learning and memory performance [7–9] and a higher susceptibility to neurodegenerative and neuropsychiatric disorders including dementia [10, 11].

Whereas a general relationship between BDNF, learning and plasticity is well documented, details such as the exact behavioral domains affected by the genetic polymorphism are less well understood. A review of 82 studies found that only 49% of the included studies demonstrated a significant association between BDNF polymorphism and cognitive performance [12]. The association was highest in the memory domain and the executive domain, lower in the attention / concentration domain, and absent in the verbal fluency domain [12]. Interestingly, while the met-allele is often associated with globally decreased performance, this review demonstrated that val/val homozygotes performed better in the memory domain but val/met carriers were superior in the executive domain [12].

Creativity is a cognitive domain often associated with unconventional ideas, innovation and progress. Creativity includes two steps: the production of original responses to a given problem, and the selection of useful solutions among the produced responses [13]. Creativity is related to cognitive flexibility, i.e. the ability to find different appropriate solutions to a problem and to flexibly choose the best solution. Cognitive flexibility has emerged as a risk and perpetuating factor for mental illness [14, 15] and increasing cognitive flexibility is a treatment goal of psychotherapy [16].

The relation between BDNF genotype and creativity, assessed with the Barrow Welsh Art Scale, was investigated in a sample of 66 patients with bipolar I disorder of whom 41 were in a manic and 25 were in a depressive episode, and 78 healthy control subjects [17]. Here, an advantage of the val/val homozygotic genotype was observed only in patients in a manic state, but not in those in a depressive state and healthy controls [17]. Apart from this study, research into the relationship between BDNF genotype and creativity is scarce.

The aim of the present work was to further analyze the relationship between BDNF genotype and creativity in a sample of healthy young students. To this end, we used the compound remote associate task (CRA), measuring verbal associative thinking as an aspect of creativity [18]. The CRA is a further developed version of the Remote Associate Task (RAT) and widely used as a measure of creativity [19]. In both tasks, three stimulus words are presented, and the participant has to find a solution word related to all three stimulus words. Whereas the solution word needs to be semantically related to the stimulus words in the RAT, the task is to build compound nouns in the CRA. A German version of the CRA has been validated by our workgroup [20]. Based on the existing literature, we hypothesized that non-met-carriers would perform better than met-carriers regarding the solution time and the number of correct solutions in the CRA.

## Methods

### Participants

The study had been approved by the local Ethics Committee of the University of Freiburg, Germany (vote number 297/11). Participants were recruited between February and May 2012.

**Table 1. Characteristics of the study sample.**

|  | val-met / met-met | val-val | p-value |
|---|---|---|---|
| N | 30 | 46 |  |
| gender | 9 male | 13 male | 0.87 |
|  | 21 female | 33 female |  |
| age in years | 23.6 ± 2.5 | 23.3 ± 3.3 | 0.62 |
| SPM | 104.8 ± 10.5 | 108.7 ± 10.1 | 0.12 |
| MWT | 110.9 ± 12.3 | 112.1 ± 13.7 | 0.69 |

SPM standard progressive matrices IQ estimate; MWT Mehrfachwahl-Wortschatztest IQ estimate. P-values refer to t-test for age, SPM and MWT and to a Chi-square test for gender.

Authors who were involved in recruitment, screening and conducting the experiments had access to information that could identify individual participants during data collection. The analyzed sample consists of 76 of 80 university students who participated in the validation study for 130 German CRA items [20]. All participants were right-handed and indicated German as their native language. Participants were free of internistic, neurological or psychiatric disorders and free of CNS-active medication. They were recruited through postings at the University of Freiburg, gave their written informed consent and received 20 Euros as a compensation.

Four participants were excluded from this sample for the present analysis because their BDNF genotype was not determined. Our sample consisted of 22 males and 54 females with a mean age of 23.4 ± 2.9 years. The BDNF genotype was val-val in 46 individuals and val-met or met-met in 30 individuals. Of the met-carriers, 27 were val-met and 3 where met-met genotypes. Met-carriers and non-met-carriers did not differ significantly regarding their age and estimated general intelligence (Table 1).

## Compound Remote Associate task (CRA)

The Compound Remote Associate task (CRA) is a measure of verbal associative thinking that is considered an aspect of creativity [18]. A German version of the CRA has been validated by our workgroup [20]. For each CRA item, the participant receives three stimulus nouns. The task is to find a solution noun that can be combined with each of the three stimulus nouns to build compound nouns as quickly as possible. An example is "battle–corn–work" as stimulus nouns, where the solution word would be 'field' (compound words "battlefield", "cornfield", "fieldwork"). Every CRA item has only one solution word.

In the present study, two non-overlapping lists consisting of 65 items each with matched item difficulty were used. Each participant received 65 CRA items with different difficulty levels.

Prior to the task, a standardized printed instruction was provided. Three example CRA items were presented and potential questions were answered prior to the actual test trial. The task was presented on a computer screen using the Presentation® software. At the beginning of each trial, a black screen appeared. Each trial had to be started by the participant by pressing the SPACE button. The three stimulus nouns of each CRA item appeared in the middle of the screen (Arial, font size 50 pt, black on white screen, separated by dashes). Participants were instructed to press the SPACE button as soon as they knew the answer and give the solution word verbally as quickly as possible. Trained study staff noted down the answer and whether or not it was correct. Participants were allowed a maximal solution time of 60 seconds per item. After they had completed half of the items, they had a recovery break of 10 minutes. The

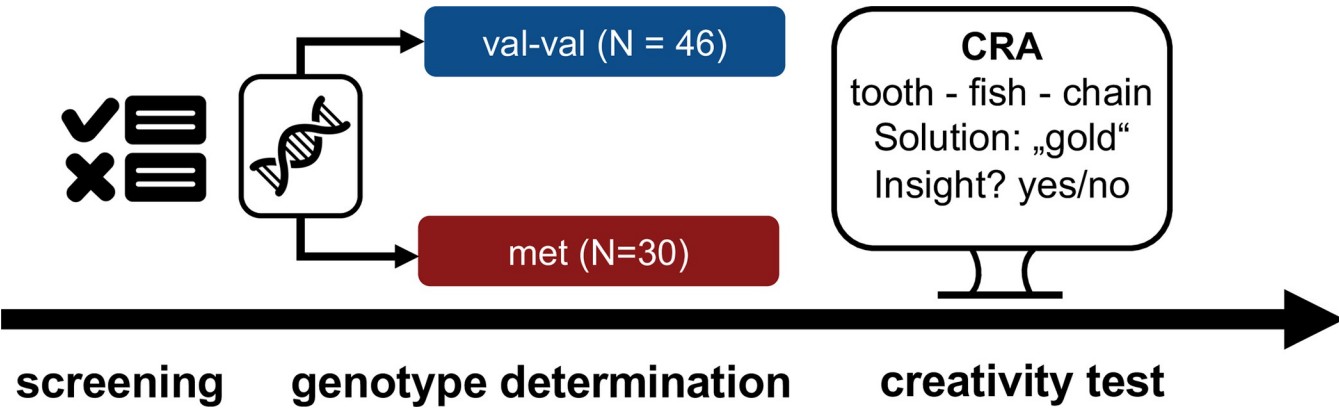

**Fig 1. Study design.**

CRA has two primary outcome variables: the number of correctly solved items per participant and the solution time for correctly solved items, i.e. the time until the space button is pressed.

### Determination of BDNF genotype

Blood sample analyses were performed by GATC Biotech in Konstanz, Germany. Genomic DNA was purified from 3 ml of whole blood and used to amplify a 281 bp polymerase chain reaction product surrounding the site of the Val66Met polymorphism for subsequent direct sequencing. Direct sequencing was performed with 3.2 pmol of the reverse primer used for initial PCR amplification. The genotype of each participant was determined following two independent rounds of direct sequencing. The following primers were used: `50-CAGGTGAGAA GAGTGATGACCA-30` (forward) and `50-GCATCACCCTGGACGTGTAC-30` (reversed). Each participant was typed as val-val, val-met or met-met. For all further analyses, met-carriers were combined to one group consisting of both the val-met and met-met genotype to allow for comparable sizes of the subgroups. The study design is depicted in Fig 1.

### Control variables

To check whether potential group differences in CRA performance were attributable to group differences in general intelligence, two control tasks were performed. The Mehrfachwahl-Wortschatztest-B (MWT-B) was used as an estimate of verbal intelligence [21]. It consists of 37 items with increasing difficulty, each consisting of a list of four nonsense words and one real word. The task is to identify the real word. Raven's standard progressive matrices (SPM) were used as an additional non-verbal measure of intelligence [22]. Both the MWT-B and the SPM allow for estimates of general intelligence.

### Statistical analyses

The sample size of 80 participants was chosen because assuming an α-error probability of 5% and a type 2 error probability of 20%, a sample size of 80 participants would be sufficient to detect a moderate effect (Cohen's d = 0.65) between groups.

The total sample of 130 CRA items (65 items per participant) was divided into three subgroups according to their level of difficulty. The level of difficulty was operationalized as the percentage of participants who correctly solved the corresponding item in the present sample. The same approach for estimating the difficulty of CRA items was used in the German validation study [20]. Items solved by up to one third of participants were classified as difficult, items

solved by up to two thirds were classified as medium, and items solved by more than two thirds were classified as easy. There were 45 items in the easy category, 42 items in the medium category and 43 items in the difficult category.

All analyses were performed with the statistical software R Studio Version 1.3.959 [23].

First, data were visually inspected for outliers using histograms. No outliers were observed. Descriptive values are given as means and standard deviations, if not indicated otherwise. We calculated analyses of variance (ANOVA) with the between-subject factor BDNF genotype (val-val vs. val-met / met-met) and the within-subject factor difficulty of the CRA item (easy, medium, difficult) as main analyses. Two ANOVA were performed with two different dependent variables: the mean time in seconds until a correct answer was given (solution time), and the number of correctly solved items. T-tests for independent samples were performed for secondary analyses. The level of significance was set at $p < .05$ (two-sided). Effect sizes were computed as partial eta square ($\eta2$) for ANOVA and as Cohen's d for t-tests. $\eta2$ from 0.01 and smaller than 0.06 indicate a small effect, $\eta2$ from 0.06 and smaller than 0.14 indicate a medium effect and $\eta2$ from 0.14 indicate a large effect. Cohen's d from 0.2 and smaller than 0.5 indicate a small effect, Cohen's d from 0.5 and smaller than 0.8 indicate a medium effect, and Cohen's d from 0.8 indicate a large effect.

## Results

### Main analysis

For the ANOVA with the solution time as a dependent variable, the BDNF genotype as a between-subject factor and the item difficulty as a within-subject factor, the main effect of the factor BDNF genotype was not significant (F = 1.66, df = 1, p = .20). The interaction between BDNF genotype and item difficulty was also not significant (F = 1.87, df = 2, p = .16). $\eta2$ indicated small effect sizes (0.02 for the main effect and 0.03 for the interaction effect). For the ANOVA with the number of correct solutions as a dependent variable, the BDNF genotype as a between-subject factor and the item difficulty as a within-subject factor, the main effect of the factor BDNF genotype was not significant (F = 2.30, df = 1, p = 0.13). The interaction between BDNF genotype and item difficulty was also not significant (F = 0.18, df = 2, p = .83). $\eta2$ indicated small effects (0.03 for the main effect and < 0.01 for the interaction).

### Secondary analyses

**Item difficulty.** As exploratory analyses, t-tests for group differences (met-carriers versus non-met-carriers) were performed within each difficulty category for the solution time and the number of correct solutions (Figs 2 and 3). Val-val genotypes had a significantly faster solution time for correctly solved items of medium difficulty (t = 2.39, df = 41.41, p = .021, Cohen's d = 0.63 indicating a medium effect size). There was no significant group difference in the solution time for correctly solved easy items (p = .25) and difficult items (p = .90). The number of correct solutions did not differ between groups for easy items (p = .52), items of medium difficulty (p = .43) and difficult items (p = .38).

## Discussion

Our study is among the first to analyze the association between BDNF polymorphism and creativity. Our main result is that for the investigated total sample of 130 CRA items, there was no significant difference between val-val carriers and val-met / met-met carriers regarding the solution time and the total number of correct solutions. We found a faster solution time in val-val carriers for items of medium difficulty. This pattern of results suggests that val-val carriers

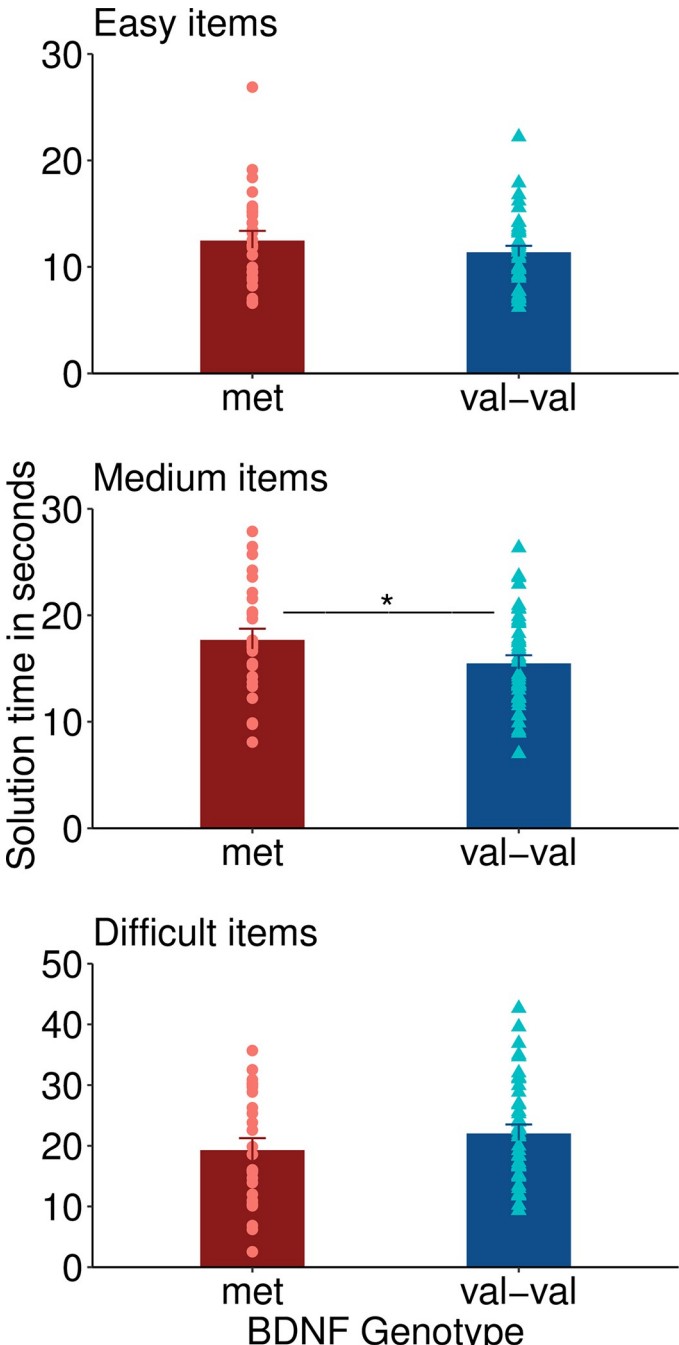

**Fig 2. Solution time.**

are generally not more creative than met-carriers, but may have an advantage when it comes to solution speed especially for items of medium difficulty. Our result is well in line with one previous study that did not find a difference in creativity, measured with a different task, in healthy subjects with and without the met allele [17].

The wider literature, however, suggests a relatively clear picture of the val-met / met-met variants leading to reduced use-dependent release of BDNF [24], decreased experience-dependent plasticity [25], abnormal hippocampal functioning [26, 27], and poorer neurocognitive

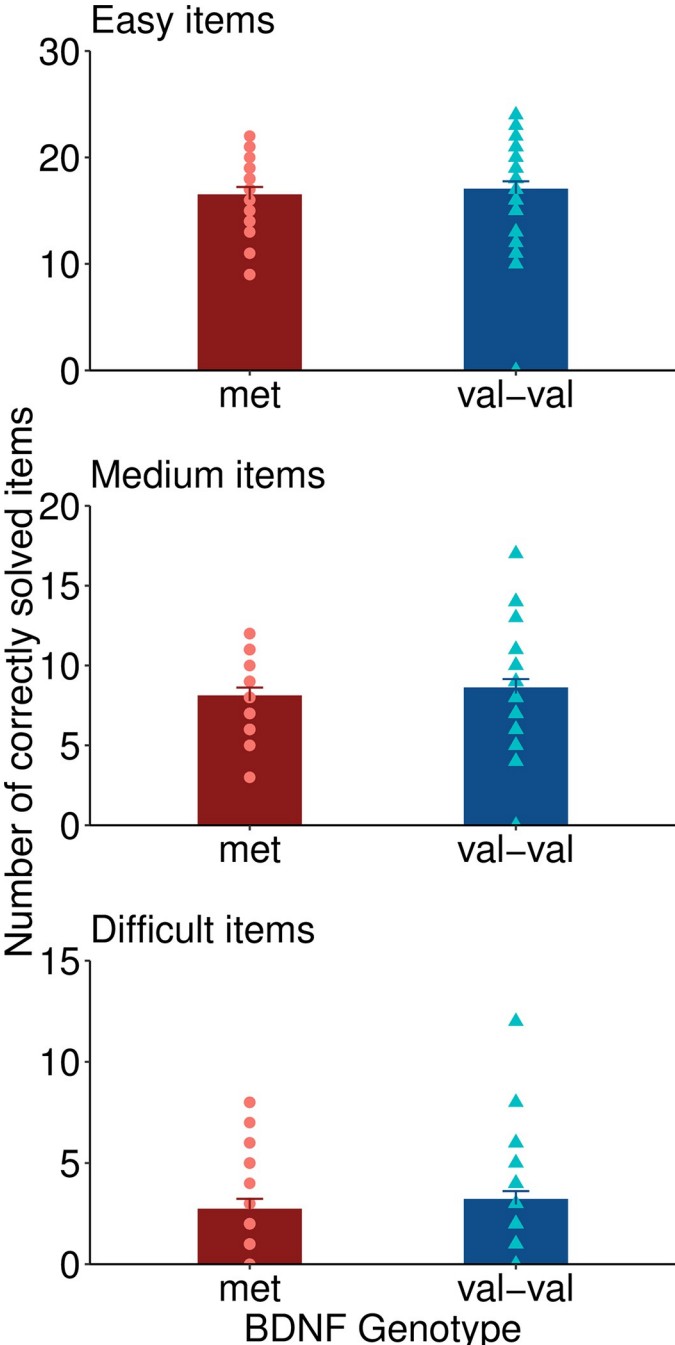

**Fig 3. Number of correct solutions.**

performance, especially in the memory domain [28]. Met-allele carriers have, on average, a steeper decline of cognitive functions during healthy aging and disease-related loss of neurons [29–31].

In contrast to this alleged common knowledge, another large cohort study of middle-aged and older adults did not find any differences in hippocampal volume and memory performance between met-carriers and non-met-carriers [32]. Many psychiatric disorders have been associated with impaired synaptic plasticity and abnormal hippocampal functioning. However,

the val-met / met-met allele has not been identified as a risk factor for depression [33, 34], post-traumatic stress disorder [35, 36], or obsessive-compulsive disorder [37]. Met-carriers appear to have a higher risk for bipolar disorder in European but not Asian populations [34]. For schizophrenia, a higher risk was found for met-met but not val-met carriers [38]. Together, the literature on the role of the BDNF polymorphism is controversial and leaves many unresolved issues. It is, overall, consistent with the notion that the Val66Met polymorphism is relevant, but not determinant, as a risk factor for neuropsychiatric disorders [39].

In neurocognitive studies investigating behavioral performance of humans with different BDNF genotypes, several factors could explain inconsistent findings: cross-sectional vs. longitudinal studies, the exact cognitive domain / brain region involved in the investigated task, and the current mental state of the investigated participants. Our study adds task-difficulty to these factors.

Recent literature suggests that better performance of val-val carriers could be limited to cross-sectional studies, whereby met-carriers seem to be superior when it comes to training effects in longitudinal studies. In healthy young individuals comparable to our sample, detrimental effects of the val-met / met-met alleles in a vocabulary learning task could be overcome by a repetitive training paradigm whereby met-carries even showed better training effects than non-met-carriers [40]. In a sample of male Vietnam combat veterans with traumatic brain injury, the met allele was a predictor of better recovery of executive functioning [41]. In our sample of students, ongoing "cognitive training" during university studies, or training effects of the trial period prior to the actual assessment, could have minimized differences that may have been present at an earlier point of time.

Previous literature has demonstrated that the effect of the BDNF polymorphism depends on the cognitive domain, whereby the val-val variant is associated with benefits in hippocampus-dependent declarative memory, but met-carriers perform better in executive tasks [12]. Also in other cognitive domains including response inhibition [42] and reasoning skills [43], met-carriers demonstrated better performance than non-met-carriers. Other domains such as verbal fluency and attention seem to be relatively unaffected by the BDNF polymorphism [12]. A potential explanation for our findings is therefore that associative thinking as a subdomain of creativity is one of the cognitive domains not largely affected by the BDNF polymorphism. A limitation of our study is that creativity is a multifaceted construct [44, 45], whereby the current study only investigated verbal associative thinking as one aspect of creativity, but may not be generalizable to other aspects. Another limitation is that due to the relatively small sample size, the statistical power in our study may have been too low to identify the effects of a single nucleotide polymorphism since such effects are known to be small. Future studies are needed to replicate our findings.

Interestingly, one previous study investigating the relationship between creativity and BDNF polymorphism suggests that the current mental state may moderate this relationship since the authors found a superiority of non-met-carriers with bipolar disorder only in a manic but not in a depressed state [17]. The same study did not find a difference in creativity in healthy controls.

In our data, we find relatively weak support for the notion that the val-val homozygote participants performed better in items with medium difficulty since our main analysis was not statistically significant. The significant finding for moderately difficult items may be attributable to the fact that variance is highest in items of medium difficulty, whereas variance is low in very easy items (solved by most subjects) and very difficult items (not solved by most subjects).

Together, the literature suggests that it would be an oversimplification to assume overall worse cognitive performance in carriers of the met-allele. On the contrary, the two genotypes may each come with different strengths and weaknesses. The prevalence of the different BDNF

genotypes differs between ethnical groups, with a prevalence of met-carriers of around 20% in European samples, but met-carriers and non-met-carriers being approximately equally distributed in Asian samples [46]. From an evolutionary standpoint, it makes little sense to assume that a genetic variant with such a high prevalence comes with no advantages and is detrimental under all circumstances [47]. Future research might identify a more refined profile of cognitive impairments and advantages of met-carriers. Understanding exactly in which domain, under which circumstances met-carriers have disadvantages or advantages could help us understand why met-carriers may have a higher risk for neuropsychiatric diseases such as dementia and may also help to inform preventive strategies and treatment options.

## Author Contributions

**Conceptualization:** Elisabeth Hertenstein, Marion Kuhn, Jonathan-Gabriel Maier, Lukas Frase, Dieter Riemann, Bernd Feige, Christoph Nissen.

**Data curation:** Bernd Feige.

**Formal analysis:** Elisabeth Hertenstein, Kristoffer Daniel Fehér, Bernd Feige.

**Investigation:** Elisabeth Hertenstein, Marion Kuhn, Nina Landmann, Jonathan-Gabriel Maier, Lukas Frase, Bernd Feige, Christoph Nissen.

**Methodology:** Elisabeth Hertenstein, Marion Kuhn, Nina Landmann, Jonathan-Gabriel Maier, Carlotta Louisa Schneider, Lukas Frase, Bernd Feige.

**Project administration:** Marion Kuhn, Nina Landmann.

**Resources:** Christoph Nissen.

**Software:** Bernd Feige.

**Supervision:** Lukas Frase, Dieter Riemann, Bernd Feige, Christoph Nissen.

**Visualization:** Elisabeth Hertenstein, Kristoffer Daniel Fehér, Christoph Nissen.

**Writing – original draft:** Elisabeth Hertenstein.

**Writing – review & editing:** Marion Kuhn, Nina Landmann, Jonathan-Gabriel Maier, Carlotta Louisa Schneider, Kristoffer Daniel Fehér, Lukas Frase, Dieter Riemann, Bernd Feige, Christoph Nissen.

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
