## [Decision Letter · Decision Letter 0]

3 May 2023

PONE-D-23-07754Brain-derived neurotrophic factor genetic polymorphism and creativityPLOS ONE

Dear Dr. Hertenstein,

Thank you for submitting your manuscript to PLOS ONE. After careful consideration, we feel that it has merit but does not fully meet PLOS ONE’s publication criteria as it currently stands. Therefore, we invite you to submit a revised version of the manuscript that addresses the points raised during the review process.

Please submit your revised manuscript by May, 30th, 2023. If you will need more time than this to complete your revisions, please reply to this message or contact the journal office at plosone@plos.org. Please include the following items when submitting your revised manuscript:A rebuttal letter that responds to each point raised by the academic editor and reviewer(s). You should upload this letter as a separate file labeled 'Response to Reviewers'.A marked-up copy of your manuscript that highlights changes made to the original version. You should upload this as a separate file labeled 'Revised Manuscript with Track Changes'.An unmarked version of your revised paper without tracked changes. You should upload this as a separate file labeled 'Manuscript'.

We look forward to receiving your revised manuscript.

Kind regards,

Amina Nasri

Academic Editor

PLOS ONE

Journal Requirements:

- https://www.frontiersin.org/articles/10.3389/fnagi.2015.00107/full

https://onlinelibrary.wiley.com/doi/full/10.1002/brb3.1009

In your revision ensure you cite all your sources (including your own works), and quote or rephrase any duplicated text outside the methods section. Further consideration is dependent on these concerns being addressed."""

3. ‘Please include your tables as part of your main manuscript and remove the individual files. Please note that supplementary tables (should remain/ be uploaded) as separate "supporting information" files

4. We note that you have stated that you will provide repository information for your data at acceptance. Should your manuscript be accepted for publication, we will hold it until you provide the relevant accession numbers or DOIs necessary to access your data. If you wish to make changes to your Data Availability statement, please describe these changes in your cover letter and we will update your Data Availability statement to reflect the information you provide

Reviewers' comments:

Reviewer's Responses to Questions

**Comments to the Author**

1. Is the manuscript technically sound, and do the data support the conclusions?

Reviewer #1: Yes

Reviewer #2: No

2. Has the statistical analysis been performed appropriately and rigorously? 

Reviewer #1: Yes

Reviewer #2: No

3. Have the authors made all data underlying the findings in their manuscript fully available?

Reviewer #1: Yes

Reviewer #2: Yes

4. Is the manuscript presented in an intelligible fashion and written in standard English?

Reviewer #1: Yes

Reviewer #2: Yes

5. Review Comments to the Author

Reviewer #1: The Manuscript “Brain-derived neurotrophic factor genetic polymorphism and creativity” reports potential implications of the BDNF Val66met polymorphism on parameters of creativity – taken by the Compound Remote Associate (CRA) task – in health young adults.

BDNF is a neurotrophin whose gene is common to all mammals and possible all invertebrates that have nervous system; whereas the Val66met polymorphism only occurs in humans. Descriptive data have shown a higher prevalence of the BDNF polymorphism in subpopulations in psychopathological disorders, such as depression, compared to general populations; as opposed to a lack of a direct effect of the BDNF val66met polymorphism on neurodegenerative-related processes.

With that been said, in my opinion this study bring up an interesting perspective for the research on the BDNF val66met polymorphism.

English language requires no revision (from a non-native speaker perspective).

The Methods and Statistics were accurate and are well described and applied.

I suggest that the authors create a Figure – Study’s design describing the methods in a Timeline/schematics to help the readers visualizing the procedures, evaluation tools (and controls) adopted.

Results appropriately presented. I know that the number of homozygote individuals for the Met-containing allele is very small and that they appear mostly always combined to the heterozygotes in the studies. However, I have to ask whether have the authors tested running the stats with the groups spared by homozygous vs heterozygous. It has been recently demonstrated an effect of the Val66met polymorphism on BDNF gene expression function in human (de Assis GG et al. 2021), and results led me to put it on perspective.

I believe the Discussion is clear and unbiased, and provides a didactic synthesis of what was pinpointed in the intersection of two major fields - behavioral Neuroscience and Molecular neurobiology. I miss a some updated references though, when it comes to the intra-cellular mechanisms demonstrably affected by the BDNF val66met polymorphism up to date (de Assis GG & Hoffman JR, 2022).

I appreciate learning from the text.

Reviewer #2: Although this is an innovative study assessing the impact of BDNF polymorphism on creativity, I have some major concerns:

1) Even though the authors provided a sample size calculation, they assumed that the polymorphism has a medium effect size, which is hard to believe. Common SNPs are known to have small effect sizes. Moreover, gene candidate studies are obsolete, and most of them are not being replicated in recent studies and GWAS.

2) In many situations, the authors refer to Val and Met alleles as different polymorphisms. They are two variants/alleles of one polymorphism

3) Page 4: Introduction: "...was investigated in a sample of 41 patients with bipolar I disorder of whom 41 were in a manic and 25 in a depressive episode". This sentence should be changed to "...was investigated in a sample of 66 patients with bipolar I disorder of whom 41 were in a manic and 25 were in a depressive episode."

4) The authors did not show if there was any difference in sex proportions between Met carriers and non-Met carriers. It should also be included in Table 1. In fact, Table 1 and sample descriptives should be moved to the Results section instead of Methods.

5) In the Discussion, the authors affirmed that Met allele frequencies vary depending on the population. However, the frequencies provided should be the ones found in known databases, such as 1000 Genomes (https://www.ncbi.nlm.nih.gov/snp/rs6265). Moreover, the term "Caucasians" should be changed to "European".

6) The bar plots presented in Figure 1 do not show the variation. Boxplots or violin plots would be more appropriate.

7) The authors should refer to the reference SNP number, rs6265.

6. PLOS authors have the option to publish the peer review history of their article (what does this mean?). If published, this will include your full peer review and any attached files.

Reviewer #1: **Yes: **Gilmara Gomes de Assis

Reviewer #2: No

---

## [Author Response · Author response to Decision Letter 0]

5 Jul 2023

Thank you very much for the opportunity to revise our manuscript. Please find enclosed our point-by-point responses to the comments of the academic editor, reviewer 1 and reviewer 2. 

Comments of the Editor: 

Thank you, we went through the documents and made sure that our manuscript meets the requirements. 

- https://www.frontiersin.org/articles/10.3389/fnagi.2015.00107/full

https://onlinelibrary.wiley.com/doi/full/10.1002/brb3.1009

In your revision ensure you cite all your sources (including your own works), and quote or rephrase any duplicated text outside the methods section. Further consideration is dependent on these concerns being addressed.

Thank you for pointing this out. We checked our manuscript for overlap with the two mentioned articles with the help of copyleaks.com. The only substantial overlap we find is for the reference section and a passage describing the BDNF rs6265 polymorphism in the introduction (textbook-knowledge). This cannot be avoided. 

We have read the Toh et al. review, summarize its results in the introduction and cite it in our reference section. The Puri et al. publication is not directly linked to our research, we had not read it before and therefore do not cite it. 

There may be overlapping text that we are not aware of – in this case please point out the passages that need to be changed. 

3. ‘Please include your tables as part of your main manuscript and remove the individual files. Please note that supplementary tables (should remain/ be uploaded) as separate "supporting information" files.

Thank you, the table has been added to the main document. 

Our initial plan was to upload the data file on Dryad. However, we have now been in contact with the person responsible for data protection at the University of Freiburg, Germany, where the data have been collected. The person has made us aware that data upload on Dryad is not allowed because the data contains potentially identifying information. We have therefore changed our statement to the following and hope for your understanding: 

“Upon consultation with the person responsible for data protection at the University of Freiburg, Germany, our data contain potentially identifying and sensitive information.

Data requests may be sent to the local ethics committee of the University of Freiburg, Germany.”

Reviewer 1

The Manuscript “Brain-derived neurotrophic factor genetic polymorphism and creativity” reports potential implications of the BDNF Val66met polymorphism on parameters of creativity – taken by the Compound Remote Associate (CRA) task – in health young adults.

BDNF is a neurotrophin whose gene is common to all mammals and possible all invertebrates that have nervous system; whereas the Val66met polymorphism only occurs in humans. Descriptive data have shown a higher prevalence of the BDNF polymorphism in subpopulations in psychopathological disorders, such as depression, compared to general populations; as opposed to a lack of a direct effect of the BDNF val66met polymorphism on neurodegenerative-related processes.

With that been said, in my opinion this study bring up an interesting perspective for the research on the BDNF val66met polymorphism.

English language requires no revision (from a non-native speaker perspective).

The Methods and Statistics were accurate and are well described and applied.

Thank you very much for your support and overall positive evaluation of our manuscript! 

I suggest that the authors create a Figure – Study’s design describing the methods in a Timeline/schematics to help the readers visualizing the procedures, evaluation tools (and controls) adopted.

Thank you for this suggestion. We have created a new figure visualizing the study design (new Figure 1). 

Results appropriately presented. I know that the number of homozygote individuals for the Met-containing allele is very small and that they appear mostly always combined to the heterozygotes in the studies. However, I have to ask whether have the authors tested running the stats with the groups spared by homozygous vs heterozygous. It has been recently demonstrated an effect of the Val66met polymorphism on BDNF gene expression function in human (de Assis GG et al. 2021), and results led me to put it on perspective.

Thank you very much for this suggestion. We appreciate this idea. However, we have not run the statistics with three groups (val-val, val-met, met-met) due to the very small sample size in the met-met group. There are only three individuals in our met-met group. To run statistics for three groups, we would have to run an Anova with group sizes of n=46, n=27 and n=3. Such an uneven group distribution and a group size below five would not fulfill the requirements for an Anova and results would not be interpretable. 

I believe the Discussion is clear and unbiased, and provides a didactic synthesis of what was pinpointed in the intersection of two major fields - behavioral Neuroscience and Molecular neurobiology. I miss a some updated references though, when it comes to the intra-cellular mechanisms demonstrably affected by the BDNF val66met polymorphism up to date (de Assis GG & Hoffman JR, 2022).

The suggested references have been added to the discussion. 

Reviewer #2: Although this is an innovative study assessing the impact of BDNF polymorphism on creativity, I have some major concerns:

1) Even though the authors provided a sample size calculation, they assumed that the polymorphism has a medium effect size, which is hard to believe. Common SNPs are known to have small effect sizes. Moreover, gene candidate studies are obsolete, and most of them are not being replicated in recent studies and GWAS.

We appreciate your criticism in terms of our sample size calculation and the problem of research results not being replicated. We are now outlining this issue in the discussion section.

2) In many situations, the authors refer to Val and Met alleles as different polymorphisms. They are two variants/alleles of one polymorphism

Thank you very much for pointing this out. We apologize for the mistake and have adapted the phrase accordingly. 

3) Page 4: Introduction: "...was investigated in a sample of 41 patients with bipolar I disorder of whom 41 were in a manic and 25 in a depressive episode". This sentence should be changed to "...was investigated in a sample of 66 patients with bipolar I disorder of whom 41 were in a manic and 25 were in a depressive episode."

Thank you very much, this has been corrected. 

4) The authors did not show if there was any difference in sex proportions between Met carriers and non-Met carriers. It should also be included in Table 1. In fact, Table 1 and sample descriptives should be moved to the Results section instead of Methods.

We apologize for this omission. The p-value for the sex distribution between Met-carriers and non-Met-carriers was .87 in a Chi square test. This has been added to table 1. 

5) In the Discussion, the authors affirmed that Met allele frequencies vary depending on the population. However, the frequencies provided should be the ones found in known databases, such as 1000 Genomes (https://www.ncbi.nlm.nih.gov/snp/rs6265). Moreover, the term "Caucasians" should be changed to "European".

Thank you, this has been adapted. 

6) The bar plots presented in Figure 1 do not show the variation. Boxplots or violin plots would be more appropriate.

Thank you for this suggestion. We agree that it is important to show variation appropriately. However, both boxplots and violin plots display the median but our statistics refer to the mean. Therefore, we chose to combine our barplots with dots for the individual data points. We hope the reviewer agrees with this choice. 

7) The authors should refer to the reference SNP number, rs6265.

Thank you, this has been added accordingly.

---

## [Decision Letter · Decision Letter 1]

29 Aug 2023

Brain-derived neurotrophic factor genetic polymorphism rs6265 and creativity

PONE-D-23-07754R1

Dear Dr. Heryenstein,

We’re pleased to inform you that your manuscript has been judged scientifically suitable for publication and will be formally accepted for publication once it meets all outstanding technical requirements.

Kind regards,

Amina Nasri

Academic Editor

PLOS ONE

Reviewers' comments:

Reviewer's Responses to Questions

**Comments to the Author**

1. If the authors have adequately addressed your comments raised in a previous round of review and you feel that this manuscript is now acceptable for publication, you may indicate that here to bypass the “Comments to the Author” section, enter your conflict of interest statement in the “Confidential to Editor” section, and submit your "Accept" recommendation.

Reviewer #1: All comments have been addressed

Reviewer #2: All comments have been addressed

2. Is the manuscript technically sound, and do the data support the conclusions?

Reviewer #1: (No Response)

Reviewer #2: Yes

3. Has the statistical analysis been performed appropriately and rigorously? 

Reviewer #1: (No Response)

Reviewer #2: Yes

4. Have the authors made all data underlying the findings in their manuscript fully available?

Reviewer #1: (No Response)

Reviewer #2: Yes

5. Is the manuscript presented in an intelligible fashion and written in standard English?

Reviewer #1: (No Response)

Reviewer #2: Yes

6. Review Comments to the Author

Reviewer #1: (No Response)

Reviewer #2: The authors have adequately addressed all of my concerns, and I do not have any further suggestions to make at this time.

7. PLOS authors have the option to publish the peer review history of their article (what does this mean?). If published, this will include your full peer review and any attached files.

Reviewer #1: **Yes: **Gilmara Gomes de Assis

Reviewer #2: No

---

## [Editor Report · Acceptance letter]

5 Sep 2023

PONE-D-23-07754R1 

Brain-derived neurotrophic factor genetic polymorphism rs6265 and creativity 

Dear Dr. Hertenstein:

I'm pleased to inform you that your manuscript has been deemed suitable for publication in PLOS ONE. Congratulations! Your manuscript is now with our production department. 

Kind regards, 

on behalf of

Dr. Amina Nasri 

Academic Editor

PLOS ONE